# A PISA-2015 Comparative Meta-Analysis between Singapore and Finland: Relations of Students’ Interest in Science, Perceived ICT Competence, and Environmental Awareness and Optimism

**DOI:** 10.3390/ijerph16245157

**Published:** 2019-12-17

**Authors:** Pei-Yi Lin, Ching Sing Chai, Morris Siu-Yung Jong

**Affiliations:** Department of Curriculum and Instruction & Centre for Learning Sciences and Technology, The Chinese University of Hong Kong, Hong Kong, Chinamjong@cuhk.edu.hk (M.S.-Y.J.)

**Keywords:** science interest, ICT competence, environmental awareness, environmental optimism, PISA

## Abstract

The aim of the present study is twofold: (1) to identify a factor structure between variables-interest in broad science topics, perceived information and communications technology (ICT) competence, environmental awareness and optimism; and (2) to explore the relations between these variables at the country level. The first part of the aim is addressed using exploratory factor analysis with data from the Program for International Student Assessment (PISA) for 15-year-old students from Singapore and Finland. The results show that a comparable structure with four factors was verified in both countries. Correlation analyses and linear regression were used to address the second part of the aim. The results show that adolescents’ interest in broad science topics can predict perceived ICT competence. Their interest in broad science topics and perceived ICT competence can predict environmental awareness in both countries. However, there is difference in predicting environmental optimism. Singaporean students’ interest in broad science topics and their perceived ICT competences are positive predictors, whereas environmental awareness is a negative predictor. Finnish students’ environmental awareness negatively predicted environmental optimism.

## 1. Introduction

Environmental issues are one of the primary concerns for future living. The United Nations Educational, Scientific and Cultural Organization (UNESCO) has been promoting the importance of environmental education for a sustainable future for more than four decades [1]. The concepts of environmental education have shifted from early focuses on ecosystems and forms of pollution to increasing recognition of sustainable development [2]. How advances in technological development and understanding of scientific knowledge are coordinated to foster awareness of the environment to bring us a greener future is a key to sustainable development [3]. This study compares the data associated with teenagers’ perceived technology competence, science interest, environmental awareness, and optimism from two of the top-performing countries in the Programme for International Student Assessment (PISA). The research aims to evaluate how top-performing PISA countries are faring in educating the next generations for sustainable living.

In the context of contemporary education, integration of science and technology in teaching and learning is key to enhancing learners’ 21st century competences [4]. While science provides the essential knowledge for people to address emerging environmental issues, information and communications technology (ICT) provide wide access to updated scientific knowledge, cognitive tools to model the problem and solution, and collaborative platforms for joint endeavors [5]. In other words, environmental issues are pedagogically valuable topics for educators to employ as authentic problems for students to deepen their science knowledge with the use of ICT. However, whether students’ science knowledge and ICT competence support their environmental awareness and hence their outlook of the future has rarely been explored at the national level [6]. Comparative studies at the national level based on big and robust data could be informative in terms of providing insights into how different education systems are dealing with the issue of sustainable development. 

### 1.1. Emerging Trends in Science Education to Promote Environmental Awareness 

Given the current emphases of science education, research is emerging to provide insights into learners’ understanding of scientific phenomena and knowledge related to lived experiences [7]. Studies have shown a growing trend in formal and informal science learning to demonstrate a greater emphasis on science and the environment [8]. The development of citizen science (CS) research; science technology society (STS); socio-scientific issues (SSI); and science, technology, society, and environment (STSE) education, has successfully situated environmental issues in the context of school science. The results of this research reported that awareness was evoked by students’ interest in science and scientific investigations. With new technologies, students can achieve both in expanding the potential for science research and in solving environmental problems [9,10]. 

In recent decades, raising concerns about the state of the natural environment have been receiving attention. In the early 1970s, environmental education was incorporated into school curricula and was presented as an interdisciplinary approach embedded in other subjects (e.g., science subjects, geography, social studies) [3,11,12]. Studies have pointed out that science knowledge is able to address environmental problems and provide solutions [13]. Moreover, people’s reflections on the co-existence and relationship between human beings and the environment are being promoted, drawing more attention to environmental awareness and more responsible behaviors [14]. Environmental awareness means the capacity to perceive, understand, and interpret the present dynamics and potential of the environment; thus, people can take action to protect, maintain, or improve the current status of the environment [15]. Broadly speaking, it is not only scientists’ work to study environmental issues and suggest solutions, but also everyone’s duty to be aware of the environmental issues, understand the causes of the problems and make efforts to protect the environment [16]. People who possess scientific literacy are more likely to have a holistic understanding of and better access to the resources to be informed about the environment [17]. In environmental education, environmental awareness could enhance students’ intention to engage in resolving pressing environmental issues. 

PISA 2006 investigated participants’ awareness of environmental and resource issues [18]. In PISA 2015, assessment of environmental literacy was developed, which includes environmental awareness and environmental optimism [19]. Susongko and Afrizal (2018) explored and found a significant relationship between adolescents’ science competencies and attitudes correlated with environmental literacy (awareness and optimism) [16]. A broadening knowledge base of environmental education has also been pointed out to foster learners’ interest in science, to develop their capacity to think about environmental situations, and to take actions to sustain the environment [20]. 

### 1.2. ICT Competence and Environmental Awareness

The relationship between ICT and the environment is a competing relationship when it is not well managed [21]. Technology advancements could be a major contributor to environment destruction. As technology advances, human ability to fish the ocean, produce plastics that are not decomposable, and extract minerals out of the earth are obvious examples that illustrate how technology could destroy the environment. ICT, as a form of technology, is also contributing to environmental issues when ICT devices are not recycled [22]. Nonetheless, any savvy teenagers are likely to be aware that current advancements place much emphasis on inventing green and clean technologies [23]. 

Emerging technologies have the potential to support activity-based, problem-based instruction, and to increase students’ capacities to collect scientific knowledge, monitor the changes in the environment, and become aware of the environment [24,25]. In the classroom, ICT is an important technological tool to support students in mapping out the multiple interconnected causal relationships that crisscross environmental issues. For example, Crawford, Holder, and O’Connor (2017) engaged learners in using mobile technologies to connect to the natural environment, and found that it can foster their environmental identity [26]. Ramasundaram, Grunwald, Mangeot, Comerford, and Bliss (2005) pointed out that learners who studied environmental issues in a simulation environment had opportunities to be involved in scientific investigations [27]. Kamarainen et al. (2013) combined an augmented reality application and environmental probes and found that it was able to enhance learners’ ecosystem science knowledge and skills, and their valuation of environmental monitoring [28]. In the field of education, studies of ICT-based instruction have generally indicated positive effects on learning [29]. The topic of sustainable development has been strongly supported by ICT and integrated into higher education in recent years [30]. Yet few studies have explored whether secondary students’ ICT competence would lead to a better understanding of the environment. While the relationship between uses of ICT, literacies, and academic achievement has been the focus of many studies using PISA and Trends in International Mathematics and Science Study (TIMSS) data, researchers have apparently not drawn on PISA data to study the relationship between contemporary teenagers’ ICT competence and their environmental awareness. We argue that this should be investigated to provide robust grounding to inform future science, technology, and environmental education practices. 

In other words, with the rapid development of scientific knowledge and new technologies, there are ample opportunities for the integration of these subject matters to be anchored in environmental issues situated in everyday life. Learners can actively participate in the problem-solving processes to construct actionable ideas pertaining to sustainable development [31]. It is pedagogically beneficial to associate science and environment as an interdisciplinary subject and to integrate technology to help learners become active participants in environmental issues [12,17]. In this paper, we used data from PISA 2015, and hypothesized that adolescent students’ interest in science topics and ICT skills may predict their awareness of environmental challenges. 

Students’ interest in science topics is critical for understanding past, current, and future environmental issues [12]. ICT application could also contribute to reducing environmental issues such as the reduction of paper usage through e-learning and e-government [32]. With the advent of technology, ICT provides opportunities to build scientific knowledge and to take action to address environmental challenges (e.g., technology-supported inquiry-based, interdisciplinary learning) [20]. This paper aims to map out the potential relationships of students’ interest in science topics, perceived ICT competence and environmental awareness to predict the environmental optimism of future living. 

## 2. Contexts in Singapore and Finland

The Environmental Performance Index (EPI) provides a quantitative basis covering environmental health and ecosystem vitality for 180 countries. Finland ranks 10th whereas Singapore ranks 49th in the 2018 EPI (see: https://epi.envirocenter.yale.edu/epi-report2018). Finland represents a Nordic education system and Singapore represents typical Asian education. They were selected for comparison in this study. Singapore and Finland are both high-performance countries in international assessment such as TIMSS and PISA. In PISA 2015, Singaporean and Finnish students ranked top of the science assessment. Both the Singaporean and Finnish curricula show an emphasis on aims for the integration of science-related, ICT-based knowledge, skills, attitudes, and values. An introduction to Singaporean and Finnish educational systems will allow for a better understanding of both countries.

### 2.1. Singaporean Context

Singapore is an island nation with a dense population. It has evolved into a highly competitive country with new ways of working and new technologies. Singapore reflects high-quality educational outcomes that rank first in science, mathematics, and reading performance as seen in the PISA results [33]. 

Singapore is renowned as a Garden City with remarkable green spaces and infrastructure [34]. Nonetheless, the country faces environmental issues such as limited land space, water, and waste problems. The emerging challenge of balancing economic imperatives with the need to conserve the natural environment needs to be addressed. Therefore, sustainable development is viewed as a top goal to be achieved [35]. Regarding knowledge of the environment, it is integrated into various subjects such as social studies, geography, moral education and the sciences as a cross-curriculum topic. It is taught through experiential learning methods to promote students’ awareness [36]. The intention of environmental education in schools is progressive, and tailored with science education with the aim of acquiring knowledge (e.g., impacts of pollution and global warming, deforestation and environmental conservation in science), values (e.g., understanding of biological ecosystems in Singapore, the responsibilities of Singaporean citizens in social studies) and action (solutions to achieve environmental sustainability in geography, science and social studies) (see: https://www.moe.gov.sg/news/parliamentary-replies/environmental-education). 

Regarding ICT and learning, Singapore has national plans for integrating and implementing technology coordinated with curricula, pedagogy, assessment, teacher professional development, and school reform. Research also provides evidence that ICT is an effective approach for supporting students’ science learning and environmental projects [37,38].

### 2.2. Finnish Context

The Finnish education system is also known as a successful system that is highly ranked in international assessments, especially for literacy [39]. In Finland, sustainable development has been brought into mainstream policies since 1993, and the integration of environmental issues in education has been extended through new learning curricula and teachers’ training. The Finnish National Core Curriculum for Basic Education pointed out that environmental issues are studied across the curriculum in science subjects, such as in biology and geography, to reach the goal of education for sustainable development (ESD) [31]. Finland is surrounded by natural resources and has a long-standing commitment to support and promote sustainable development. The sustainable development element has been one of the seven topics emphasized in the national core curriculum for basic education since 2006. Since 2013, approximately 40% of schools have had a sustainable development plan in place [40]. Students learn to understand the interrelation between human beings and the natural environment, and raise awareness of the environment and commitment to building a sustainable future [41]. 

Increasingly, environmental education is performed in various forms of integrated curricula and pedagogies (e.g., project-based collaboration; see, for example, [41]; inquiry-based learning; see, for example, [42]). Therefore, students have opportunities to learn about complex and real-world problems related to the environment within and outside of school science subjects [43]. They are also encouraged to share and create scientific knowledge that paves the way for sustainable development [41]. 

Studies have also shed light on the uses of ICT in recent science and environmental education [24,44]. The Finnish National Core Curriculum indicates that “competence in information and communication technology” and “participation, involvement and building a sustainable future” are transversal competences in the Finnish National Core Curriculum (see: https://www.oph.fi/sites/default/files/documents/new-national-core-curriculum-for-basic-education.pdf). ICT education in Finland is guided by a clear vision of how the availability of ICT-related skills and capacities could facilitate learning and concerns of environmental issues [24]. The cross-curricular projects not only emphasized social competencies such as collaboration, but also characterized innovative ICT skills to engage learners in active and independent learning to search for, organize, and analyze information, and to communicate and propose their ideas in various media forms [44]. 

In the present study, we considered four factors assessed in PISA 2015, including students’ interest in broad science topics, perceived ICT competence, environmental awareness, and environmental optimism. The research questions are: Can a comparable factor structure of interest in broad science topics, perceived ICT competence, environmental awareness and optimism be established between Singapore and Finland?Is there a significant difference between Singapore and Finland students’ scores for interest in broad science topics, perceived ICT competence, environmental awareness and optimism?What is the relationship between interest in broad science topics, perceived ICT competence, environmental awareness, and optimism in Singapore and Finland?

## 3. Method

### 3.1. Participants

The sample of this study employed data publicly released from the PISA 2015 database. PISA is the triennial survey of adolescent students around the world launched by the Organization for Economic Co-operation and Development (OECD). PISA 2015 consisted of 72 participating countries and regions. The sample of this study is 15-year-old students in Singapore and Finland. The Singapore PISA 2015 sample is composed of 6115 students (48.6% female and 51.4% male) and the Finland sample is composed of 5882 students (48.7% female and 51.3% male). This study included student-level factors (see items in Table 1), that is, students’ interest in broad science topics, perceived ICT competence, environmental awareness, and environmental optimism. 

### 3.2. Variables

PISA provides a reliable and well-developed assessment administered by the OECD since 2000. It is conducted across the globe to understand 15-year-old students’ literacies in reading, mathematics and science. Several studies have used PISA 2015 data to gain insights into various aspects of students’ achievement and literacies [45,46,47]. PISA 2015 focused on science literacy, and the data collection also includes ICT uses and environmental literacy (awareness and optimism). Environmental awareness indicates a person’s particular knowledge and sensitivity related to the global environment and its causes [48]. Environmental optimism reflects a person’s feelings, attitudes and beliefs about environmental improvement and protection as a concern for the future [49].

All variables were taken from the student questionnaire in PISA 2015. The self-reported scales show students’ perceptions of or attitude towards their agreement with the asked questions. This study includes the following variables.

#### 3.2.1. Interest in Broad Science Topics

PISA 2015 measures students’ interest in five broad science topics, or subjects with response (i.e., “not interested,” “hardly interested,” “interested” or “highly interested”; ranging from 1 to 4) for the topics related to the biosphere (e.g. ecosystem services, sustainability); motion and forces (e.g., velocity, friction, magnetic and gravitational forces); energy and its transformation (e.g., conservation, chemical reactions); the universe and its history; and how science can help us prevent disease. The internal reliability (Cronbach’s α) values of the scale with respect to the Singapore and Finland data were 0.77 and 0.83 respectively.

#### 3.2.2. Perceived ICT Competence

A four-point Likert scale was used for perceived ICT competence (ranging from 1 to 4: strongly disagree to strongly agree). The scale includes items to understand if students feel confident about their knowledge of ICT and how to use ICT using the following four items: “I feel comfortable using digital devices that I am less familiar with”; “If my friends and relatives want to buy new digital devices or applications, I can give them advice”; “When I come across problems with digital devices, I think I can solve them”; and “If my friends and relatives have a problem with digital devices, I can help them.” The internal reliability (Cronbach’s α) values of the scale with respect to the Singapore and Finland data were 0.82 and 0.85 respectively.

#### 3.2.3. Environmental Awareness

Investigation of environmental awareness shows students’ concerns about environmental issues. The scale was obtained from the questionnaire to measure students’ views regarding the environment and its problems. They were asked to respond to the following environmental issues: “the increase in greenhouse gases in the atmosphere”; “the use of genetically modified organisms (GMO)”; “nuclear waste; the consequences of clearing forests for other land use”; “air pollution”; “extinction of plants and animals; and water shortage.” Students’ positive responses to the four-point scale (i.e., I have never heard of this; I have heard about this but I would not be able to explain what it is really about; I know something about this and could explain the general issue; I am familiar with this and I would be able to explain this well) indicate higher levels of environmental awareness. The internal reliability (Cronbach’s α) values of the scale with respect to the Singapore and Finland data were 0.86 and 0.85 respectively.

#### 3.2.4. Environmental Optimism

In PISA 2015, students were asked how they thought that the problems associated with environmental issues (as mentioned in environmental awareness) would improve or get worse over the next 20 years. Students’ expectations of environmental issues as a concern for the future were a three-point Likert scale with responses from 1 to 3 (i.e., get worse; stay about the same; improve). The internal reliability (Cronbach’s α) values of the scale with respect to the Singapore and Finland data were 0.85 and 0.81 respectively.

### 3.3. Data Analyses

The analysis of the data was conducted in accordance with the research questions of the study. Firstly, regarding research question 1, we analyzed the data using Kline’s (2005) criteria and found that values of skewness (−1.113 to 0.350) and kurtosis (−1.174 to 0.694) indicated the dataset was normally distributed (skewness and kurtosis are under |3| and |10| respectively) [50]. Exploratory factor analyses (EFA) with SPSS version 21 (IBM Corp. Released 2012. IBM SPSS Statistics for Windows, Version 21.0. Armonk, NY: IBM Corp.) were employed to examine the construct validity of the Singapore and Finland datasets. Items that address students’ attitudes towards science (e.g., interest in broad science topics), competence beliefs of technology (e.g., perceived ICT competence) and attention to environmental issues (e.g., environmental awareness and optimism) were included. 

Regarding research question 2, descriptive statistics were calculated to depict levels of students’ attention to environmental issues, their willingness to engage in the science topics and their confidence in using technology. Next, the independent sample *t*-test was conducted to compare differences between the means in the Singapore and Finland datasets. 

As for the third research question, Pearson’s correlation analysis and regression analyses were conducted to reflect the correlations among factors and to ascertain the predictive relationships between the factors. 

## 4. Results

The results are presented in the following three sections. First, a measurement of students’ related variables in interest in broad science topics, perceived ICT competence, environmental awareness, and environmental optimism was established. Second, associations among the four factors were shown. Third, regression models were conducted to examine the predictions of environmental variables and perceived ICT competence. 

### 4.1. Establishment of a Model

The descriptive statistics for all variables used are given in Table 1. EFA was applied to examine the factors in the measurement. The principal component factoring method was used in the factor analysis. The direct oblimin method was employed to produce an optimal structure. Factors were identified by factor loadings and Eigen values (exceeding the value of 1). In this study, items of environmental awareness, environmental optimism, interest in broad science topics, and perceived ICT competence were accepted as items in the students’ questionnaire in PISA 2015. One item “I feel comfortable using my digital devices at home” was deleted because its extraction community was 0.187 in the Singapore dataset. 

In order to determine the applicability of factor analysis, the Kaiser–Meyer–Olkin (KMO) Value and Bartlett’s test of sphericity were calculated before the EFA. KMO values over 0.50 (KMO = 0.85 in the Singapore dataset; KMO = 0.85 in the Finland dataset) indicated that factor analysis sampling was appropriate. Bartlett’s test of sphericity was found to be significant and adequate for EFA (*X^2^* = 44,428.78, df = 253, *p* < 0.01 in the Singapore dataset; *X^2^* =38,545.77, df = 253, *p* < 0.01 in the Finland dataset). The factor loading of the items ranged from 0.44 to 0.89 in the Singapore dataset and 0.36 to 0.89 in the Finland dataset. Total explained variance was found to be 57.07% in the Singapore dataset and 55.61% in the Finland dataset. 

In Table 2, the means and standard deviations for latent variables measuring students’ interest in broad science topics, perceived ICT competence, environmental awareness and optimism in Singapore, Finland, and across the OECD countries are presented. In general, Singaporean students have high levels of interest in broad science topics and revealed higher levels of environmental awareness than the OECD average scores and those of the Finnish students. Both the Singaporean and Finnish students perceived lower levels of ICT competence and environmental optimism than the OECD average scores. Further analysis was conducted using independent samples *t*-tests (Singaporean versus Finnish students) and showed that students in Singapore significantly outperformed students in Finland in interest in broad science topics, perceived ICT competence and environmental awareness. However, there were no significant differences (*p* > 0.05) between Singapore and Finland students’ environmental optimism.

### 4.2. Relationships among Variables

The correlations among the latent variables are given in Table 3. Separate correlations were computed for the Singaporean and Finnish students. Interest in broad science topics, perceived ICT competence and environmental awareness were positively and significantly correlated. Particularly, interest in broad science topics was more strongly and positively correlated with environmental awareness in both countries. Regarding environmental awareness and environmental optimism, they were negatively and significantly correlated. However, there are differences between Singapore and Finland. In Singapore, students’ environmental optimism was positively and significantly correlated with perceived ICT competence, whereas Finnish students’ environmental optimism was negatively and significantly correlated with interest in broad science topics. The low (i.e., all < 0.441) and significant correlations between the dependent and independent variables is the indicator of the applicability regression analysis.

### 4.3. Regression Analyses among Variables

In regression analyses, we separated students into groups by country, to examine the different extents to which a set of explanatory variables account for the variance in students’ perceptions of environmental optimism (dependent variable) in PISA 2015. Table 4, Table 5 and Table 6 present the standardized coefficients of the regression models predicting environmental optimism and awareness, and perceived ICT competence. In Table 4, we have only the explanatory variable (i.e., interest in broad science topics) predicting students’ perceived ICT competence. The linear regression models in both countries were significant, accounting for 1.7–2.8% of the variance. In Table 5, we have explanatory variables (i.e., interest in broad science topics and perceived ICT competence) predicting students’ environmental awareness. The multiple regression models in both countries were significant, accounting for 16.9–19.5% of the variance. In Table 6, we have explanatory variables (i.e., interest in broad science topics, perceived ICT competence, and students’ environmental awareness) measuring students’ optimism. The results revealed that Singaporean students’ interest in broad science topics and perceived ICT competence are significantly positive predictors, whereas environmental awareness was a significantly negative predictor. In Finland, environmental awareness was the only significantly negative predictor. Both models accounted for 1.9–2.7% of the variance. None of the variance inflation factors (VIF) exceed in the abovementioned models.

## 5. Discussion

Given the importance of environmental issues in today’s world, many school practitioners and researchers analyze and assess students’ perceptions of environmental issues. Our study proposed that interest in science-related topics and perceived ICT competence are critical for understanding past, current, and potential future environmental issues, particularly the human-caused environmental problems. We analyzed data collected in the PISA 2015 assessment. This study brings students’ interest in broad science topics, perceived ICT competence, environmental awareness, and environmental optimism together. 

First, we were able to identify a comparable factor structure to compare the two countries. This indicates that future researchers who are interested in understanding the relationship between science interest, ICT competence, environmental awareness, and optimism could use the PISA data to understand how various countries are leveraging these factors to promote environmental education. There should perhaps be more comparative studies, especially when the countries’ environmental education has been well articulated and implemented. We chose Singapore and Finland as two contrasting high performing countries with different cultural roots, and our findings indicate that there are some differences. 

Second, with regard to the differences, *t*-tests revealed that, compared to Finnish students, the Singaporean students had higher levels of interest in broad science topics, perceived better ICT competence, and had more environmental awareness. On the other hand, this study highlighted that Singapore and Finland did not differ significantly in terms of environmental optimism. These differences seem to reflect the general differences between the two countries in the achievement scores. Singapore as a small nation is placing much emphasis on science, and ICT in education [36] and it is facing obvious environmental threats due to rising sea level and haze problems from neighboring countries. 

Next, the Pearson’s correlation analyses showed that the associations of the students’ interest in broad science topics, perceived competence in ICT uses and environmental awareness were positive in both countries. However, environmental awareness and optimism were negatively associated. Results also showed that Singaporean students’ perceptions of ICT competence were positively associated with environmental optimism, indicating that they had a higher degree of familiarity with ICT uses, and this could contribute to more optimistic outlook about improvements in these environmental issues. The underlying belief the result reflects could be that technology is a solution to environmental problems. In Finland, students had a higher degree of interest in broad science topics and were negative about environmental improvements. This could imply that students may believe that science is insufficient to resolve environmental issues. In either case, further qualitative research is needed to further unpack the relationships unearth in this study. 

According to the results obtained from the multiple linear regression analyses, both countries reported that students’ interest in broad science topics predicted their competence in ICT. Moreover, students who were characterized as having interest in broad science topics, as well as possessing ICT competence, manifested concerns about the environmental issues. Regarding environmental optimism, Singaporean students’ interest in broad science topics and perceived ICT competence were positive predictors, whereas environmental awareness was a negative predictor. This is aligned with studies indicating that increasing interest in scientific knowledge and the uses of ICT could provide directions to successfully resolve environmental problems [51]. For example, the capabilities of ICTs and other related technologies can be employed to identify environmental problems, collect environmental information, and find solutions to environmental problems. In the Finland data, environmental awareness was the only negative predictor of environmental optimism. It is worth noting that in Finland, students gain scientific knowledge based on craft activities rather than technology. Although research has pointed out that Finnish students were being motivated to learn how to protect the environment with the help of technology [52], there is still a considerable amount of work to do in developing a more technology-integrated learning environment. 

## 6. Conclusions and Limitations

Engaging youth as environmental citizens and developing their ability to identify the global environmental issues have been promoted [53]. Since PISA 2006, there have been investigations of students’ environmental awareness and responsibility. This study provides robust evidence that advocating a sense of environmental awareness and optimism were supported by students’ positive interest and engagement in science and ICT competences. However, further studies on exploring the relationship between science interest, ICT uses, and environmental literacy are needed. In parallel, teachers’ environmental literacy and training should also be considered [54]. 

The findings in the present study showed that the prediction of Singaporean students’ environmental optimism was mediated through positive students’ interest in broad science topics and ICT competences. They may be more ready to use their science knowledge to understand environmental issues and design solutions with the help of ICT. Results from PISA 2015 indicated that Singaporean students outperformed in science, mathematics, and reading assessment. The present study further explored and provided evidence that Singaporean students are able to discover broad scientific interest in the context of massive information flows. They are also able to use ICT and have developed a greater understanding of environmental literacies. 

Although results show that Finnish students’ interest in science and perceived ICT competence did not predict their environmental optimism, new Finnish curriculum reforms from August 2016 have introduced multi-disciplinary learning and competences such as multi-literacy and ICT, and are working to develop a better integration of science, technology and environmental education (see: https://www.oph.fi/sites/default/files/documents/new-national-core-curriculum-for-basic-education.pdf). 

In this study, we examined factors that influence students’ environmental awareness and optimism with students from Singapore and Finland. The aim is to contribute to the promotion of environmental awareness among students in education. However, there are some limitations to the present study that should be noted. For one, the PISA assessment is not a curriculum survey to assess students’ learning outcomes. Students report self-evaluations of their learning, motivation, interests, and perceptions related to science, technology, and the environment. The interpretation of PISA data should be more cautious. Next, the purpose of this study is to provide insights into more understanding of adolescents’ science interest, ICT competence and environmental literacies in two countries, but its application in various other countries needs further study. Third, the secondary data analysis is limited to what is taught in school classes and might not be well-connected to national policies. This study identifies science as the foundational discipline that predicts Singapore and Finland students’ environmental literacy. In addition, students’ ICT confidence could also influence their environmental literacy. Given the ubiquitous access to ICT, it is likely to be a good means to promote students’ environmental literacy. Our findings, however, indicate that there seems to be room for improvement in promoting environmental education for students and future generations.

Lastly, some researchers, as pointed out by one of the reviewers for this study, hold the view that PISA data should be treated as qualitative variables since the Likert scale was employed to elicit students’ responses. Consequently, they would suggest that the appropriate statistical analyses should be non-parametric method such as Mann–Whitney U test and Spearman instead of Pearson correlations. We conducted the Mann–Whitney U test and Spearman correlations as appended in Appendix B and Appendix C for the readers’ reference. The notable changes as indicated are that the non-significant finding in environmental optimism (see Appendix B is related to Table A2) has become significantly different and that the positively significant correlations among Singaporean students between perceived ICT competence and environmental optimism is reversed as significantly negative (see Appendix C in relation to Table A3). Overall, using a non-parametric test may result in slightly different findings and it can be an alternative way to explore the PISA data. However, such differences would require discussion of methodological issues that is beyond the focus of this study.

## Figures and Tables

**Table 1 ijerph-16-05157-t001:** Means, SD values, Cronbach’s alpha if item deleted, and EFA factor loadings for each item.

	Singapore				Finland			
Items	M (SD)	Alpha if Item Deleted	Factor loadings	% Variance	M (SD)	Alpha if Item Deleted	Factor loadings	% Variance
1. Interest in broad science topics				8.02				7.95
ST095Q04NA	2.46 (0.97)	0.73	0.70		2.02 (0.86)	0.80	0.72	
ST095Q07NA	2.59 (0.97)	0.73	0.75		2.35 (0.96)	0.78	0.86	
ST095Q08NA	2.68 (0.96)	0.70	0.81		2.34 (0.97)	0.77	0.88	
ST095Q13NA	3.05 (1.01)	0.75	0.64		2.84 (1.03)	0.82	0.64	
ST095Q15NA	3.11 (0.91)	0.74	0.66		2.78 (0.97)	0.81	0.68	
2. Perceived ICT Competence				10.88				11.39
IC014Q03NA	2.68 (0.78)	0.85	0.66		2.44 (0.78)	0.87	0.70	
IC014Q04NA	2.76 (0.80)	0.77	0.81		2.88 (0.74)	0.79	0.85	
IC014Q08NA	2.95 (0.74)	0.74	0.86		2.88 (0.75)	0.79	0.86	
IC014Q09NA	2.84 (0.79)	0.72	0.89		2.86 (0.77)	0.77	0.89	
3. Environmental Awareness				22.29				22.18
ST092Q01TA	3.17 (0.84)	0.83	0.79		2.95 (0.81)	0.83	0.67	
ST092Q02TA	2.32 (1.02)	0.87	0.55		2.05 (0.85)	0.86	0.44	
ST092Q04TA	2.45 (0.83)	0.85	0.61		2.78 (0.75)	0.83	0.70	
ST092Q05TA	3.36 (0.78)	0.83	0.85		2.90 (0.80)	0.82	0.76	
ST092Q06TA	3.44 (0.69)	0.83	0.84		3.22 (0.67)	0.82	0.84	
ST092Q08TA	3.19 (0.79)	0.83	0.81		3.14 (0.73)	0.82	0.84	
ST092Q09TA	3.12 (0.80)	0.83	0.80		2.86 (0.81)	0.84	0.75	
4. Environmental Optimism				15.88				14.09
ST093Q01TA	1.51 (0.75)	0.83	0.77		1.46 (0.69)	0.78	0.73	
ST093Q03TA	1.50 (0.71)	0.83	0.76		1.71 (0.67)	0.78	0.68	
ST093Q04TA	1.46 (0.72)	0.82	0.77		1.53 (0.66)	0.78	0.68	
ST093Q05TA	1.80 (0.81)	0.83	0.70		1.77 (0.70)	0.79	0.64	
ST093Q06TA	1.73 (0.74)	0.83	0.70		1.63 (0.69)	0.79	0.65	
ST093Q07TA	1.54 (0.77)	0.82	0.76		1.46 (0.67)	0.77	0.74	
ST093Q08TA	1.88 (0.75)	0.85	0.60		1.77 (0.60)	0.79	0.61	

**Table 2 ijerph-16-05157-t002:** Description of latent variables in Singapore, Finland, and the OECD countries, and independent *t*-test results for Singapore and Finland.

Variable	OECD Countries M (SD) (*n* = 235937)	Singapore M (SD) (*n* = 4586)	Finland M (SD) (*n* = 4312)	*t*-Value
1. Interest in broad science topics	2.63 (0.74)	2.77 (0.70)	2.45 (0.74)	−21.624 ***
2. Perceived ICT competence	2.85 (0.66)	2.81 (0.63)	2.77 (0.63)	−3.596 ***
3. Environmental awareness	2.86 (0.65)	3.01 (0.61)	2.85 (0.56)	−14.994 ***
4. Environmental optimism	1.68 (0.60)	1.63 (0.55)	1.62 (0.46)	−1.279

Note 1: *** *p* < 0.001.

**Table 3 ijerph-16-05157-t003:** Correlation coefficients between latent variables in Singapore and Finland.

Variable	1	2	3	4
1. Interest in broad science topics		0.166 ***	0.441 ***	−0.035 *
2. Perceived ICT competence	0.131 ***		0.125 ***	0.015
3. Environmental awareness	0.407 ***	0.123 ***		−0.151 ***
4. Environmental optimism	−0.019	0.045 **	−0.147 ***	

Note 1: * *p* < 0.05; ** *p* < 0.01; *** *p* < 0.001, Note 2: In correlation analyses: the lower triangle is the Singapore data; the upper triangle is the Finland data.

**Table 4 ijerph-16-05157-t004:** Standardized (β) coefficients of the simple linear regression models for predicting perceived ICT competence.

	Singapore (*n* = 4815)	Finland (*n* = 4640)
Interest in broad science topics	0.131 ***	0.166 ***
*R* ^2^	0.017	0.028
*F*-value	84.284	131.538
VIF	1.000	1.000

Note 1: *** *p* < 0.001

**Table 5 ijerph-16-05157-t005:** Standardized (β) coefficients of the multiple linear regression models for predicting environmental awareness.

	Singapore (*n* = 0 4685)	Finland (*n* = 0 4494)
Interest in broad science topics	0.396 ***	0.433 ***
Perceived ICT competence	0.070 ***	0.040 **
*R* ^2^	0.169	0.195
*F*-value	475.438	542.985
Max VIF	1.017 (Science interest and ICT competence)	1.029 (Science interest and ICT competence)

Note 1: ** *p* < 0.01; *** *p* < 0.001

**Table 6 ijerph-16-05157-t006:** Standardized (β) coefficients of the multiple linear regression models for predicting environmental optimism.

	Singapore (*n* = 4586)	Finland (*n* = 4312)
Interest in broad science topics	0.046 **	0.031
Perceived ICT competence	0.063 ***	0.027
Environmental awareness	−0.171 ***	−0.149 ***
*R* ^2^	0.027	0.019
*F*-value	42.676	27.957
Max VIF	1.209 (Science interest)	1.267(Science interest)

Note 1: ** *p* < 0.01; *** *p* < 0.001

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
