# Peer review of "A PISA-2015 Comparative Meta-Analysis between Singapore and Finland: Relations of Students’ Interest in Science, Perceived ICT Competence, and Environmental Awareness and Optimism"

_ijerph, 2019, doi:10.3390/ijerph16245157_

Round 1
Reviewer 1 Report
An adequate and correct manuscript with a very interesting subject matter is presented. However, no citation from 2019 has been added, and I recommend that authors add references from this year. Moreover, the references are not adapted to the format of the journal, I suggest you consult its rules.
On the other hand, it would be advisable to include references from the journal IJERPH. Some articles on environmental awareness are published:
https://www.mdpi.com/1660-4601/16/3/362
https://www.mdpi.com/1660-4601/16/6/905
Reviewer 2 Report
This work is based on a large dataset. It is well presented, however it has a major problem.
Although qualitative variables are employed (Likert scale) for eliciting students' responses, statistical analyses uses parametric tests in the place of the appropriate non parametric equivalent ones. For example Mann Whitney U test should be used in the place of t-test. Spearman instead of Pearson correlations. Logit instead of linear regression, etc.
I suggest to revisit the statistical analysis and resubmit this work. The findings will not differ much, but is important to use the appropriate analysis.
Reviewer 3 Report
This is an interesting paper, that has been soundly presented. There are some issues that should be addressed prior to publication.
Line 46 – We don’t know, at least not with the literature presented, whether the outlook is indeed ‘positive’ so the word should probably be removed.
Line 54 needs a reference (‘Studies have shown…’)
Line 75 also needs a reference (‘People who are…’)
Line 79 – It would be useful to have a little more information here on how environmental literacy was measure in PISA 2015.
Line 79 – ‘Existing studies….’ and yet only one study is cited. This needs a reference to more studies or removal of the plural.
Line 149 – which PISA results – a reference would be useful here.
Line 189 – ‘Studies have shed light…’ this needs several references.
Line 248-249 – Internal consistency as measured by? (I assume Cronbach’s alpha, if so, this needs to be mentioned.) This should be addressed throughout the manuscript. Also note, internal consistency, or internal reliability, but not both.
Line 257 – to be consistent with previous example, please insert quotation marks.
The analysis is clearly presented and accurate. I wonder if there needs to be a post-hoc multiple comparison correction presented in this section.
The end of the paper seems abrupt – you end with the limitations, while I would have expected a few sentences, in a conclusion perhaps, that brings everything together. Please consider inserting this.
Overall, this is an interesting paper that warrants publication.
Reviewer 4 Report
The introduction was well written and the author provided sufficient background information.
The research questions were clearly identified throughout the manuscript.
Throughout the manuscript, the authors presented multiple factual claims without providing references to offer support. For example, line 53, line 72, and line 74. The authors should consider clearly defining the pronoun "it" and limit the use of the pronoun to provide clarity. The use of abbreviations, specifically, "ICT" must be defined in the abstract as readers view the abstract prior to the subsequent content.
The methodology section was not clearly presented as the methods section was difficult to follow. For example, the Analytic Strategy subheading included content that made the text hard to read. The data analysis was consistent with the research questions.
The conclusion was well written and consistent with the results that were presented.
Round 2
Reviewer 2 Report
My initial disposition cannot be changed as I see no serious effort to correct the statistical analysis. As it stands the manuscript still carries the problems I have noticed in my first report. The Authors seem not to, or do not want, understand that they have employed unsuitable statistical tests.
